# QS21-Initiated Fusion of Liposomal Small Unilamellar Vesicles to Form ALFQ Results in Concentration of Most of the Monophosphoryl Lipid A, QS21, and Cholesterol in Giant Unilamellar Vesicles

**DOI:** 10.3390/pharmaceutics15092212

**Published:** 2023-08-26

**Authors:** Erwin G. Abucayon, Mangala Rao, Gary R. Matyas, Carl R. Alving

**Affiliations:** 1U.S. Military HIV Research Program, Walter Reed Army Institute of Research, 503 Robert Grant Avenue, Silver Spring, MD 20910, USA; mrao@hivresearch.org (M.R.); gmatyas@hivresearch.org (G.R.M.); 2Henry M. Jackson Foundation for the Advancement of Military Medicine, 6720A Rockledge Drive, Bethesda, MD 20817, USA

**Keywords:** giant unilamellar vesicle, vaccine adjuvant, monophosphoryl lipid A, cholesterol, QS21, QS-21, membrane fusion, liposome

## Abstract

Army Liposome Formulation with QS21 (ALFQ), a vaccine adjuvant preparation, comprises liposomes containing saturated phospholipids, with 55 mol% cholesterol relative to the phospholipids, and two adjuvants, monophosphoryl lipid A (MPLA) and QS21 saponin. A unique feature of ALFQ is the formation of giant unilamellar vesicles (GUVs) having diameters >1.0 µm, due to a remarkable fusion event initiated during the addition of QS21 to nanoliposomes containing MPLA and 55 mol% cholesterol relative to the total phospholipids. This results in a polydisperse size distribution of ALFQ particles, with diameters ranging from ~50 nm to ~30,000 nm. The purpose of this work was to gain insights into the unique fusion reaction of nanovesicles leading to GUVs induced by QS21. This fusion reaction was probed by comparing the lipid compositions and structures of vesicles purified from ALFQ, which were >1 µm (i.e., GUVs) and the smaller vesicles with diameter <1 µm. Here, we demonstrate that after differential centrifugation, cholesterol, MPLA, and QS21 in the liposomal phospholipid bilayers were present mainly in GUVs (in the pellet). Presumably, this occurred by rapid lateral diffusion during the transition from nanosize to microsize particles. While liposomal phospholipid recoveries by weight in the pellet and supernatant were 44% and 36%, respectively, higher percentages by weight of the cholesterol (~88%), MPLA (94%), and QS21 (96%) were recovered in the pellet containing GUVs, and ≤10% of these individual liposomal constituents were recovered in the supernatant. Despite the polydispersity of ALFQ, most of the cholesterol, and almost all of the adjuvant molecules, were present in the GUVs. We hypothesize that the binding of QS21 to cholesterol caused new structural nanodomains, and subsequent interleaflet coupling in the lipid bilayer might have initiated the fusion process, leading to creation of GUVs. However, the polar regions of MPLA and QS21 together have a “sugar lawn” of ten sugars, the hydrophilicity of which might have provided a driving force for rapid lateral diffusion and concentration of the MPLA and QS21 in the GUVs.

## 1. Introduction

In the development of liposomes as models of cell membranes or as delivery vehicles, two important variables have often arisen: the overall size of the vesicles, and the number of bilayer membranes in the vesicles, e.g., unilamellar, multilamellar, or oligolamellar vesicles [1]. The sizes of unilamellar vesicles are sometimes characterized as small unilamellar vesicles (SUV) (diameter ~50–100 nm) or large unilamellar vesicles (LUV) (~100–1000 nm). For example, a reverse-phase evaporation method for encapsulation of large amounts of water-soluble materials in LUVs with a diameter range of 0.2–1 µm was created by Szoka and Papahadjopoulos [2], and a dehydration–rehydration method was devised by Gregoriadis and colleagues for manufacture of giant (5.5 µm) multilamellar liposomes [3]. The manufacture and characteristics of giant multilamellar cationic vesicles that served as adjuvant carriers was also described by Perrie and associates [4]. However, the discovery and synthesis of giant unilamellar vesicles (GUV) (spherical vesicles with diameters >1000 nm), bounded by a phospholipid lipid bilayer membrane surrounding an aqueous core, originally started as a subset of the liposome field [5,6,7,8]. GUVs are now a rapidly emerging field of cell-sized particles (or even prototypic cells) that are manufactured by various methods using amphiphilic lipids or polymers, with numerous applications because of the ability to visualize the GUV particles by light microscopy or even with the naked eye [9,10,11,12,13,14,15,16,17]. Despite the extensive literature on numerous types of GUVs for a huge number of applications, to our knowledge detailed studies on the use of GUVs as carriers of vaccine adjuvants have not been conducted.

In 2015, a novel GUV-containing liposome composition known as Army Liposome Formulation with QS21 (ALFQ) was introduced as a potent carrier of adjuvants for vaccines [18]. ALFQ consists of liposomes (with diameters in the range of 50 nm to ~30 µm), containing saturated phospholipids, dimyristoyl phosphatidylcholine (DMPC) and dimyristoyl phosphatidylglycerol (DMPG), 55 mol% cholesterol compared to the total bulk phospholipid, and two adjuvants, a saponin QS21 from the bark of the *Quillaja saponaria* tree, and native or synthetic monophosphoryl lipid A (MPLA), an anchor lipid present in Gram-negative lipopolysaccharide [18,19].

ALFQ is being extensively examined in preclinical and clinical studies as a potent adjuvant constituent of numerous vaccines. It is currently in various stages of seven current, completed, or planned clinical trials as a vaccine adjuvant, including experimental vaccines for malaria [20,21], HIV-1 [22,23,24], *Campylobacter* diarrhea [25], and COVID-19 [26], and initial studies indicate that it has few local or systemic side effects in humans [19,27]. Despite its broadly advanced initial clinical development, the chemical and physical characteristics, and the relative compositions of the GUVs in ALFQ when compared to the SUVs from which they are derived, have never been characterized. Here, we purified GUVs by differential centrifugation of ALFQ and studied the physical and chemical composition of lipid bilayers after centrifugation. The composition of adjuvants in the pellet (containing GUVs) were compared to those in the SUVs and LUVs in the supernatant. Striking differences in the lipid and adjuvant compositions were found between the SUVs, LUVs, and GUVs that suggest mechanistic insights about the fusion processes leading to GUVs.

## 2. Materials and Methods

### 2.1. Materials and Reagents

Dimyristoyl phosphatidylcholine (DMPC), dimyristoyl phosphatidylglycerol (DMPG), synthetic MPLA (3D-PHAD^®^), and cholesterol (plant derived) for liposomal preparation and working standards for analytical measurements, along with fluorescently labelled (TopFluor^®^) cholesterol, were purchased from Avanti Polar Lipids Inc., (CRODA), Alabaster, AL, USA. DMPC and cholesterol were dissolved in freshly distilled chloroform, and DMPG and MPLA were dissolved in chloroform:methanol (9:1). Methanol, water, isopropyl alcohol (IPA), dichloromethane, formic acid, and ammonium formate (all Optima^TM^ LC–MS grade) were purchased from Fisher Scientific, Asheville, NC, USA. QS21 (cGMP grade) was purchased from Desert King International, San Diego, CA, USA. The QS21 working standard was prepared from in-house HPLC-purified cGMP-manufactured QS21.

### 2.2. ALFQ

ALFQ was prepared by a lipid deposition method described previously [18,28,29]. Briefly, a precursor liposome composition (ALF55), containing DMPC and DMPG, 55 mol% cholesterol relative to both phospholipids (DMPC + DMPG), and MPLA, was incubated with QS21 to initiate the fusion reaction leading to the formation of ALFQ. ALFQ then became a polydisperse liposome suspension that consisted of DMPC, DMPG, cholesterol, MPLA, and QS21. The total phospholipids of ALFQ were 45.9 mM relative to the aqueous suspending medium, Sorensen’s phosphate-buffered saline (SPBS) pH 6.2. The molar ratios of DMPC:DMPG:Chol:MPLA:QS21 in the liposomes were 9:1:12.2:0.114:0.044. ALFQ was further diluted in SPBS to achieve 22.9 mM phospholipid relative to the SPBS.

### 2.3. AS01-Like Liposomes

Adjuvant system AS01, which is present in several Glaxo SmithKline (GSK) vaccines, was not available. Therefore, AS01-like liposomes were created to mimic some of the known major chemical and physical properties of AS01. In this context, three major differences distinguish AS01 from ALFQ: unsaturated instead of saturated liposomal phospholipids, 33% instead of 55% cholesterol, and MPLA comprising native MPL^®^ instead of synthetic 3D-PHAD^®^. To achieve a formulation like AS01 for comparison with ALFQ to the best degree possible, the AS01-like suspension consisted of dioleoyl phosphatidylcholine (DOPC) as the bulk phospholipid, 33 mol% cholesterol relative to the total phospholipid, and MPLA consisting of synthetic 3D-PHAD^®^. As described previously [29], the total concentrations of DOPC and MPLA (3D-PHAD^®^) relative to water were 2.54 mM and 0.065 mM, respectively, with 100 μg/mL of QS21. The molar ratio of liposomal DOPC:MPLA was 88:1.

### 2.4. Purification of GUVs

The ALFQ suspension was initially diluted 1:10 using SPBS. The diluted ALFQ samples (5 mL) were placed in glass centrifuge tubes and centrifuged in an RC5C Sorvall^®^ centrifuge (DuPont, Newtown, CT, USA) at 8000 RPM (767× *g*) for 10 min at 22 °C. The supernatant was decanted into a clean glass tube, and the pellet was washed 3× with SPBS to remove trapped small vesicles. Both the supernatant from the first centrifugation and the washed pellet were viewed by light microscopy for size characterization. The small vesicles in the supernatant were also analyzed by dynamic light scattering (DLS). The quantities (μg) of the liposomal phospholipids, cholesterol, MPLA, and QS21 in the washed pellet and the supernatant were determined using an ultraperformance liquid chromatography–high-resolution tandem mass spectrometry (UPLC–MS/MS) method.

### 2.5. Microscopy

Size characterization of GUVs (in the pellet) employed a bright field with 75× magnification in an Olympus BH2-RFCA microscope equipped with an Olympus DP71 camera (Waltham, MA, USA). For confocal microscopy studies, the total liposomal phospholipid concentration was ~2.29 mM (1:10 dilution), with 0.25% labelled cholesterol (TopFluor^®^) relative to phospholipid. Images were taken with an Olympus Fluoview FV1200 confocal microscope (Waltham, MA, USA) using a 60×/1.42 Oil Pan Apo. All images were processed with FV10-ASW 4.2 software.

### 2.6. Dynamic Light Scattering for Size and Zeta Potential

The weighted hydrodynamic diameter of ALF55 (1:50 dilution) was determined based on the intensity of its dynamic light scattering in SPBS on a Malvern Zetasizer Nano S (Malvern, Worcestershire, UK) equipped with a 633 nm laser. On the same instrument, the zeta potential at 25 °C was measured.

### 2.7. Quantitative Analysis of Lipid Components by UPLC–MS/MS

The quantitation of liposomal lipid constituents was performed in a Thermo Scientific Vanquish UHPLC coupled with a Q-Exactive Quadrupole-Orbitrap detector. The concentration of QS21 was determined as recently described [30]. Briefly, the separation was carried out in an Agilent Zorbax Eclipse Plus C18 column (4.6 mm ID × 50 mm, 1.8 μm particle size), using water (A) and methanol (B), acidified with 0.1% formic acid, as mobile phases with a constant flow of 0.5 mL/min at a controlled column temperature of 35 °C. The injection volume was set at 5 μL. All data were acquired using negative electrospray ionization in parallel reaction monitoring (PRM) mode. The electrospray and source settings were as follows: 2.5 kV (capillary voltage), 320 °C (capillary temperature), 25 AU (sheath gas flow rate), 10 AU (Aux gas flow rate), and 300 °C (Aux gas temperature). QS21 isomers were detected as [M − H]^−^ under the following PRM transitions: QS21 1 and QS21 2 with *m*/*z* 1987.9169 > 485.3272 at 10.91 and 10.41 min (chromatographic RT), respectively, and QS21 R1 and R2 isomers with *m*/*z* 1855.8746 > 485.3268 at 10.83 and 11.40 min, respectively. Quantification was done using an external calibration method with equal weighting scheme in TraceFinder 5.1 (ThermoScientific, Waltham, MA, USA).

The liposomal phospholipids (DMPC and DMPG) were quantified using a UPLC–MS/MS-based method. The separation was carried out in an Agilent Zorbax Eclipse Plus C18 column (4.6 mm ID × 50 mm, 1.8 μm particle size), using 95/5 methanol/water (A) and IPA (B), with 5 mM ammonium formate and 0.1% formic acid, as mobile phases with a constant flow of 0.5 mL/min at a column temperature of 45 °C. The injection volume was set at 1 μL. All data were acquired using a negative electrospray ionization in PRM mode. The electrospray and source settings were the same as those described in QS21 quantitation. DMPC and DMPG were detected using PRM transitions of *m*/*z* 772.50 > 227.20 at 5.44 min, and 665.44 > 227.20 at 3.85 min, respectively. Quantification was done using an external calibration method with 1/x weighing scheme in TraceFinder 5.1 (ThermoScientific, Waltham, MA, USA).

The concentrations of 3D-PHAD^®^ in the liposomes were determined using a UPLC–MS/MS with analytical conditions similar to that of the phospholipids. The 3D-PHAD^®^ was detected using a PRM transition of *m*/*z* 1518.08 > 1004.65 at 13.07 min. The quantification was done using an external calibration method with an equal weighing scheme.

The cholesterol concentrations were quantified by UPLC–MS/MS using the following analytical conditions. The separation was carried out in a Kinetex^®^ Phenomenex C18 column (2.1 mm ID × 150 mm, 2.6 µm particle size), using 95/5 methanol/water (A) and 62/36/2 methanol/dichloromethane/water (B), with 5 mM ammonium formate and 0.1% formic acid, as mobile phases with a constant flow of 0.4 mL/min at a column temperature of 40 °C. The injection volume was set at 2 μL. All data were acquired using a positive electrospray ionization in PRM mode. The electrospray and source settings were the same as those described in QS21 quantitation. Cholesterol was detected using a PRM transition of *m*/*z* 369.35 > 147.12 at 4.26 min. Quantification was done using an external calibration method with an equal weighing scheme in TraceFinder 5.1 (ThermoScientific, Waltham, MA, USA).

### 2.8. Data Analysis

Statistical analyses were performed in a GraphPad Prism 9.0. Comparison of the zeta potentials of different liposomes was determined using ordinary one-way ANOVA. An unpaired *t*-test with Welch’s correction was employed to compare the lipid components (by weight) of GUVs (pellet) with SUVs (supernatant). Differences among values are statistically significant if *p* < 0.05.

## 3. Results

### 3.1. Separation of GUVs from SUVs and LUVs by Centrifugation

Centrifugation was employed to separate GUVs from an ALFQ polydisperse liposomal suspension that contained nanoparticles (SUVs and LUVs with diameters <1 µm) and microparticles (GUVs with diameters >1 µm) (Figure 1). A suspension of “AS01-like” liposomes that contained only SUV particles was included for comparison. The AS01-like liposomal suspension containing unsaturated (dioleoyl) phospholipids, together with 33 mol% cholesterol and QS21, was transparent, with a marked Tyndall effect due to the presence of nanoparticles [29]. In contrast, ALFQ was characterized by an opaque white milky appearance. As shown in Figure 1, sedimentation of AS01-like liposomes at 15,000 RPM (2696× *g*) for 15 min at 22 °C did not result in the formation of a pellet. In contrast, sedimentation of ALFQ resulted in the formation of a large pellet, and the supernatant contained a slight Tyndall effect. After subsequent sedimentation and washing of the ALFQ pellet, the supernatant became clear with much less evidence of a Tyndall effect. From this, it was clear that liposomal microparticles in ALFQ could be separated from nanoparticles by centrifugation.

Images of the pellets and supernatants after centrifugation of ALFQ at different speeds revealed that the original polydisperse sample could be separated into various subfractions (Appendix A). Centrifugation of an ALFQ suspension (1:10 dilution) at 8000 RPM (767× *g*) for 10 min at 22 °C (Appendix A) resulted in the separation of GUVs (in the pellet) from nearly all the smaller vesicles (supernatant) (Appendix A). Although initial centrifugation was done at 15,000 RPM (2696× *g*), essentially the same separation was observed using an initial speed of 8000 RPM (767× *g*). In all subsequent experiments, 8000 RPM (767× *g*) was utilized, and the isolated soft pellet was washed 3× to remove any trapped smaller vesicles.

As shown by light microscopy in Figure 2A,B, the pellet was composed mainly of vesicles with diameters >1 μm, while the supernatant was characterized mainly of vesicles having diameters <1 μm. The DLS analysis, as shown in Figure 2C, confirmed that the 767× *g* supernatant exhibited modal peaks of diameters ~80 nm (SUVs) and ~550 nm (LUVs).

### 3.2. Fluorescence Analysis of GUVs

Confocal fluorescence studies of ALFQ labelled with TopFluor^®^ cholesterol showed that the pellets isolated from centrifuged fluorescent-labelled ALFQ suspension comprised unilamellar vesicles with diameters >1 μm (Figure 3A,B). In contrast, most of the vesicles visible in the supernatant had diameters <1 μm (Figure 3C,D). Giant vesicles separated by centrifugation appeared to be unilamellar (Figure 3A,B).

### 3.3. Total Lipid Recovery by Weight after Centrifugation

Quantitative analysis of the pellets and supernatants by UPLC–MS/MS was conducted to determine the ratio and distribution of their lipid components. Shown in Table 1 are the total weights and percent (%) recoveries of the lipids in the pellet and supernatant after centrifugation of ALFQ at 8000 RPM (767× *g*).

In Table 1, it is shown that after centrifugation, ~88% total lipid recovery occurred by weight, and ~63% was in the pellet, while the supernatant had only ~25%. Some losses were incurred during subsequent washing of the pellet after centrifugation. Production of GUVs resulted from cannibalization of the SUVs and LUVs that were present in ALFQ. As shown in Table 2, the total phospholipid recovery (DMPC + DMPC) by weight in the pellet was 44%, compared to 36% in the supernatant. However, much higher amounts, by weight, of cholesterol, MPLA, and QS21 were present in the pellet than in the supernatant.

### 3.4. Molar Amounts of Individual Lipids in the Pellet and Supernatant after Centrifugation

As shown in Figure 4, on a molar basis nearly all of the total QS21 and MPLA adjuvant molecules were in the pellet rather than in the supernatant.

Regardless of the liposome size, whether SUVs, LUVs, or GUVs, the total phospholipids (in this case, DMPC + DMPG) are the largest mole fraction compared to other lipid constituents in the liposomal lipid bilayer. Because of this, the mole % of cholesterol in the liposomes is often compared on a molar basis to the total phospholipids. This is particularly important in the present case because it is generally believed that QS21 specifically binds with high affinity to cholesterol. As shown in Figure 5, ALFQ contained 55 mol% cholesterol before centrifugation, but after centrifugation, the pellet contained 72 mol% cholesterol, and the supernatant contained only 25 mol% cholesterol. In the case of MPLA and QS21, their % molar ratios relative to total phospholipids were 1.0% and 0.5%, respectively, in ALFQ; 3.0% and 1.0% in the pellet; and 0.3% and 0.1% in the supernatant. Overall, the observed changes in the liposomal lipid compositions after centrifugation resulted in substantial variations of the (DMPC + DMPG):Chol:MPLA:QS21 molar ratios. The starting ALFQ has (DMPC + DMPG):Chol:MPLA:QS21 molar ratios of 1:1.27:0.01:0.005. However, after centrifugation, the resulting (DMPC + DMPG):Chol: MPLA:QS21 molar ratios of the purified GUVs in the pellet was found to be 1:2.55:0.03:0.01, while that of the SUVs in the supernatant have molar ratios of 1:0.33:0.003:0.001.

### 3.5. Zeta Potentials of Particles

The zeta potentials of the precursor ALF55, ALFQ, and purified GUVs in the pellet and SUVs in the supernatant after centrifugation of ALFQ are summarized in Table 3. Although ALF55 and ALFQ contain different amounts of lipid components, both exhibit comparable negative zeta potentials in SPBS at pH = 6.1. A more negative zeta potential (−11.43 ± 1.15 mV) was observed for the purified GUVs, in comparison with the precursor ALF55 and the starting ALFQ. On the other hand, the SUVs and LUVs in the supernatant exhibited a less negative zeta potential (−8.38 ± 0.97 mV). Based on ordinary one-way ANOVA multiple comparisons test, the observed slight decreases and increases in the zeta potentials of purified GUV and SUV relative to the starting ALFQ were not statistically significant. However, a significant difference (*p* < 0.005) was observed in the comparison of zeta potentials between GUVs and SUVs.

### 3.6. Is MPLA Required for GUV Formation?

To answer this question, ALFQ without MPLA was prepared by the addition of QS21 to ALF55 without MPLA suspended in SPBS. The starting ALF55 without MPLA exhibited a modal peak of ~50 nm with PDI ~0.2 by DLS analysis (Figure 6). As expected, the size of the starting ALF55 without MPLA was similar to that of ALF55 with MPLA. Addition of QS21 to ALF55 without MPLA resulted in the formation of polydisperse vesicles of a size in the range of 50 nm to ~30 μm, consistent with the size range observed for the original ALFQ. This observation revealed that MPLA was not required in the formation of GUV in ALFQ.

## 4. Discussion

Previous studies have demonstrated that saturation of the fatty acyl groups in the phospholipids of the SUVs from which the GUVs are derived is a primary requirement for creation of the ALFQ GUV structure [29]. Van der Waals forces in the hydrophobic regions of saturated phospholipids are greater than those in unsaturated phospholipids, and this provides sufficiently increased tensile strength to withstand the stresses on the lipid membrane during GUV formation after addition of QS21. This explains why the AS01 (GSK) adjuvant, which consists of liposomes containing DOPC, cholesterol, MPLA, and QS21, with a diameter of ~107 nm, does not form GUVs [29,31] (also see Figure 1). However, the requirement for saturated phospholipids does not by itself explain why the extraordinary fusion reaction was initiated and sustained by the addition of QS21 to the SUVs, leading to the formation of ALFQ GUVs. Although the mechanism of initiation of the fusion reaction by the addition of QS21 to ALF55 is not yet clear, it is known that saponins such as QS21 bind with high affinity to cholesterol and cause various structural changes in the cholesterol and liposomes [32,33,34], and permeability changes due to lesions in erythrocytes, leading to hemolysis. We speculate that nanodomain discontinuities and heterogeneities in ALF55 cholesterol caused by QS21 binding might have led to membrane fusion that was initiated by sudden new interleaflet coupling in the lipid bilayer [35].

To help further understand the fusion reaction leading to ALFQ GUVs, the present work extends the previously described concept that the bulk phospholipids of ALFQ serve as a kind of highway in which the lateral diffusion of cholesterol, QS21, and MPLA reach apparent equilibria that result in unique lipid raft-type constructs in the lipid bilayer of the liposomal ALFQ structure. In the present work, although the fluorescent labelling of ALFQ with TopFluor^®^ cholesterol clearly illustrated the unilamellar nature of the ALFQ GUVs, the lateral distribution of the fluorescent-labelled cholesterol was heterogeneous, thus suggesting the presence of lipid aggregations in the phospholipid bilayer (Figure 3A,B and Figure 6B,C,E,F).

Analysis of the total weights and molar amounts of each of the lipid constituents in the centrifuged pellets (GUVs) and supernatants (SUVs and LUVs) when compared with uncentrifuged ALFQ revealed startling differences in the distributions of phospholipids, cholesterol, QS21, and MPLA. First, after centrifugation, 88% of all the lipids (by weight) were recovered in the combined pellet and supernatant (Table 1). Second, as shown in Table 2, the total amount of bulk phospholipid (DMPC + DMPG) was divided, with 44% (by weight) in the pellet and 36% in the supernatant. This suggested that after adding QS21 to ALF55 to form ALFQ, slightly more than half of the large number of nanoliposomes in ALF55 were incorporated into the phospholipid bilayer of a smaller number of micron-sized GUV liposomes. In contrast, as shown in Table 2 and Figure 4, the vast majority of cholesterol, QS21, and MPLA (by weight) were recovered in the GUVs in the pellet (88%, 96%, and 94%, respectively), when compared to the SUVs and LUVs in the supernatant (10%, 7%, and 10%, respectively). Thus, although the bulk phospholipids of ALFQ were divided relatively evenly between GUVs and smaller particles, almost all of the adjuvants (on a molar basis) were present in the GUVs (in the pellet), and hardly any were in the SUVs and LUVs (in the supernatant). Therefore, even though ALFQ has a wide polydisperse size distribution (with particles having diameters generally from about 50 nm to about 30,000 nm), and although it is known as a remarkable adjuvant [19], virtually all of the adjuvant molecules in ALFQ are concentrated in the GUV particles rather than in the smaller (LUV and SUV) particles. In addition, Figure 5 reveals that the mole % of cholesterol compared to the bulk phospholipids (DMPC + DMPG) was 55 mol% in ALFQ but was 72 mol% in the pellet (GUV) and only 25% in the supernatant (SUVs and LUVs).

The changes in the lipid compositions of the purified GUVs and SUVs relative to the starting ALFQ resulted in apparent changes in their zeta potentials. The GUVs exhibited a slightly more negative zeta potential (−11.43 ± 1.15 mV) in SPBS (pH = 6.1), relative to the precursor ALF55 and ALFQ with zeta potentials of −10.50 ± 0.52 mV and −10.60 ± 0.62 mV, respectively. The opposite was observed for the SUVs, which had a less negative zeta potential (−8.38 ± 0.97 mV) as compared to those of ALF55 and ALFQ. The more negative zeta potential in the GUVs can be ascribed to the increased amount of MPLA that contributed an overall negative charge, in addition to DMPG. Consistently, the SUVs that contained a small amount of MPLA showed a less negative zeta potential. In comparison to our previously reported zeta potentials of ALF55 and polydisperse ALFQ vesicles, which were suspended in PBS (pH ~7.4) [18], the measured zeta potentials of vesicles suspended in SPBS (pH = 6.1) were relatively less negative (see Table 3). These discrepancies might have been due in part to the differences in the pH and buffer components of the suspending media [36,37].

All of the above suggest that the MPLA and QS21 adjuvants might have been the initial and continual driving forces for creation of the GUVs. This could have been due to faster lateral diffusion rates caused by the hydrophilic forces of the QS21 (with eight sugars) and the MPLA (with two sugars) when compared to the bulk phospholipids. Although both adjuvants likely contributed to the formation of the GUVs, Figure 6 demonstrates that GUVs were formed even in ALF55-type liposomes that lacked MPLA. It also seems likely that because the added QS21, which was essentially irreversibly bound to cholesterol [38], a large fraction of the cholesterol in the original ALF55 SUVs to which the added QS21 was bound was dragged with an increased lateral diffusion rate into GUVs, resulting in a huge (72%) cholesterol:phospholipid % molar ratio (Figure 5).

With respect to the hydrophilic nature of the ten sugars that are present on the surface of the two ALFQ adjuvants (QS21 and MPLA), recently it has been proposed that the sugars on the two adjuvants might form a “sugar lawn” on the surface of ALFQ [19]. It was further hypothesized that the proposed “sugar lawn” on ALFQ might be home to and bind to numerous types of cell-associated mammalian lectins and that binding to lectin receptors might confer a major portion of the vaccine adjuvant activity of ALFQ. If this is true, then the differential centrifugation methods in the present work, which localized most of the adjuvant molecules in a high concentration on the surfaces of purified GUVs from ALFQ, could result in a high potency adjuvant activity for a variety of applications. The huge concentration of the liposomal adjuvant molecules in GUVs suggests that the well-documented in vivo adjuvant activities of ALFQ were likely due to giant adjuvanted particles. Future immunization studies with antigens mixed with GUVs will be conducted in comparison with the adjuvanticity of small particles (SUVs) containing MPLA and QS21.

As reviewed in a historical perspective by Nair and Bajaj [17], the broad literature in the GUV field includes a wide variety of methods for the creation of GUVs and applications that include biophysical insights about membranes and creation of prototypic cells [17]. In this context, the present work reveals a new and novel biochemical and biophysical example of membrane fusion resulting in a GUV initiated by QS21 saponin that includes enhanced lateral diffusion in the liposomal lipid bilayer of QS21 bound both to cholesterol and MPLA. Despite the large GUV literature, we have not found a previous description of a GUV that could serve as a vaccine adjuvant. Thus, based on the present work, an important further application might be to use purified ALFQ GUVs as a vehicle to deliver encapsulated antigenic peptides or proteins [16], RNA or DNA [12,39], or even whole viruses or cells, as elegantly achieved with giant multilamellar vesicles by Gregory Gregoriadis et al. [3]. Any of these concepts as a new type of adjuvanted formulation might induce enhanced or prolonged immunity. Targeting of the ALFQ GUVs might also be achieved by including glycolipids or other glycoconjugates in addition to QS21 and MPLA, to target specific mammalian lectins [11,40,41]. In view of this, future studies on developing methods for encapsulating materials into the aqueous interior of ALFQ GUVs, and targeting of the GUVs, will be explored.

## 5. Conclusions

The ALFQ adjuvant formulation is quite complex, consisting of vesicles containing different sizes, structures, and possibly even different morphologies induced by QS21. Saturated phospholipids and cholesterol comprise the underlying liposomal structure, with small amounts of MPLA and QS21 in the lipid bilayer providing all the adjuvant molecules. Addition of QS21 to precursor SUVs containing the phospholipids, 55% cholesterol, and MPLA to form ALFQ results in a dramatic fusion reaction, resulting in generation of GUVs (>1 µm diameter) in addition to SUVs and LUVs. A major objective of this work was to examine the chemistry and physical structures of the vesicles after the fusion reaction. Purification of GUVs was achieved by differential centrifugation. It was then demonstrated that most of the MPLA and QS21, and even cholesterol, had migrated to the GUVs due to increased lateral diffusion, leaving the SUVs and LUVs with large amounts of phospholipid but only small amounts of MPLA, QS21, and cholesterol. Possible mechanisms for this remarkable result are discussed, and it is suggested that this new method of creating the purified adjuvanted GUVs could exhibit increased future potency as a vaccine adjuvant when compared to the original polydisperse ALFQ. It is also suggested that these results might enable future studies focusing on addition of targeting elements to the surface of the GUVs and even on encapsulation of large quantities of protein or peptides, RNA or DNA, and viruses, or even whole cells, within the GUVs.

## Figures and Tables

**Figure 1 pharmaceutics-15-02212-f001:**
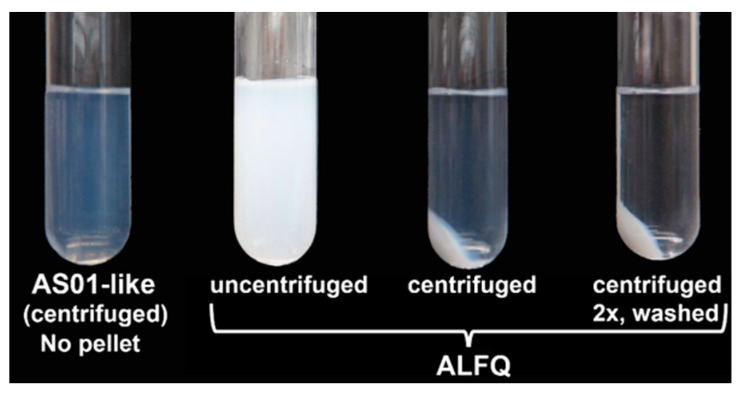
Physical properties of ALFQ in comparison with AS01-like liposomal adjuvant. Unlike AS01-like liposomes, centrifugation of ALFQ at 15,000 RPM (2696× *g*) for 15 min at 22 °C resulted in the formation of a pellet.

**Figure 2 pharmaceutics-15-02212-f002:**
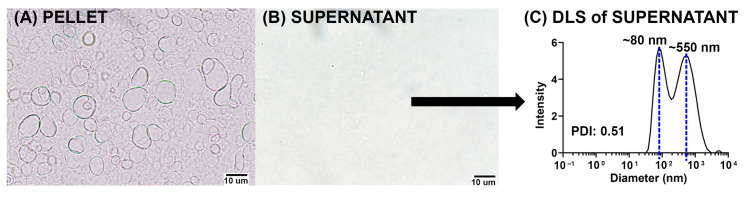
Characterization of the pellets and supernatant from ALFQ centrifugation at 8000 RPM (767× *g*). (**A**) Visualization by phase-contrast light microscopy studies showed that the pellet is made of GUVs with diameter >1 μm. (**B**) Majority of the vesicles in the supernatant have diameters <1 μm. (**C**) DLS studies also show that the modal sizes of the vesicles in the supernatant are ~80 nm (SUVs) and ~550 nm (LUVs).

**Figure 3 pharmaceutics-15-02212-f003:**
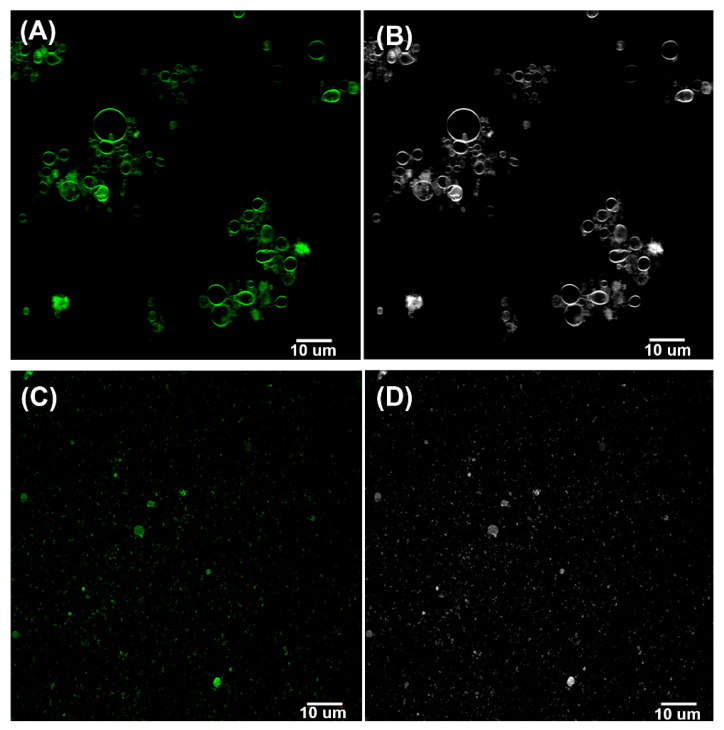
Confocal microscopy images of the pellet (**A**), with the corresponding image in gray scale (**B**), and supernatant (**C**), with the corresponding image in gray scale (**D**) after centrifugation of ALFQ labelled with TopFluor^®^ cholesterol at 8000 RPM (767× *g*) for 10 min at 22 °C. The pellet was washed 3× before conducting confocal microscopy studies.

**Figure 4 pharmaceutics-15-02212-f004:**
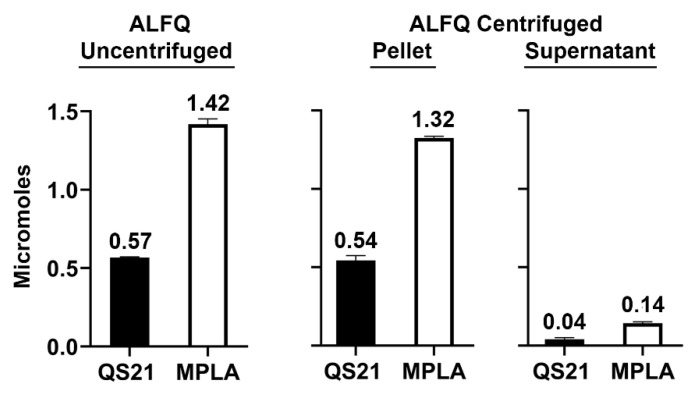
Molar amounts of QS21 and MPLA in the pellet and supernatant after centrifugation, relative to the original ALFQ.

**Figure 5 pharmaceutics-15-02212-f005:**
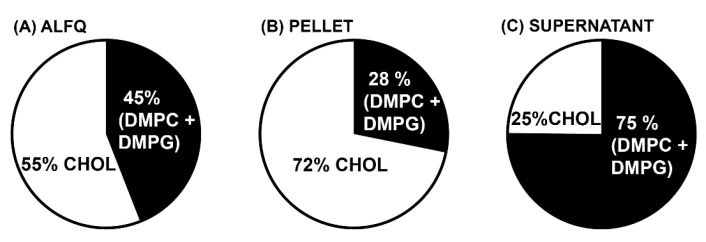
Molar ratios of cholesterol relative to total phospholipids (DMPC + DMPG) in ALFQ (**A**), centrifuged pellet (**B**), and supernatant (**C**). The pellet, consisting of GUVs, has substantially higher cholesterol content relative to total phospholipids compared to the original ALFQ.

**Figure 6 pharmaceutics-15-02212-f006:**
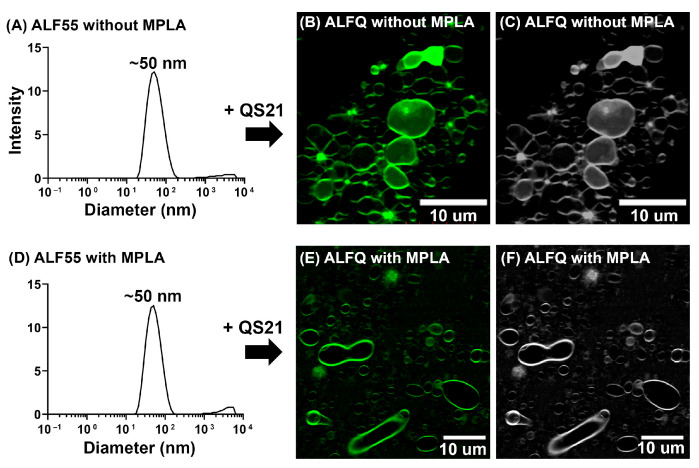
Biophysical characterization and visualization of ALF55 and ALFQ without MPLA, in comparison with the original ALFQ suspension. (**A**) DLS analysis of ALF55 without MPLA showing a modal peak of ~50 nm. (**B**) Visualization of ALFQ without MPLA using confocal microscopy, (**C**) with the corresponding grayscale image. (**D**) DLS analysis of ALF55 with MPLA showing a modal peak of ~50 nm. (**E**) Visualization of the original ALFQ suspension using confocal microscopy, (**F**) with the corresponding grayscale image.

**Table 1 pharmaceutics-15-02212-t001:** Total lipid recovery by weight in the pellet and supernatant.

Sample *	Weight (mg)	% Recovery
ALFQ	139.63	---
Pellet	88.29	63
Supernatant	34.60	25

* 5 mL of ALFQ suspension containing 22.9 mM of bulk phospholipids. After centrifugation at 8000 RPM (767× *g*), the total lipid weights in the washed (3×) pellet and the supernatant after a single centrifugation were determined by adding the individual weights of DMPC, DMPG, cholesterol, MPLA, and QS21 quantified by UPLC–MS/MS, as shown in Table 2. The % recoveries in the pellet and supernatant were calculated relative to the total weights of lipids in the original ALFQ.

**Table 2 pharmaceutics-15-02212-t002:** Recovery by weight (mg) of each ALFQ constituent after centrifugation.

Component *	Uncentrifuged ALFQ		Centrifuged ALFQ		
		Pellet	% Recovery	Supernatant	% Recovery
DMPC	71.18 ± 0.68	31.07 ± 0.27	44	25.98 ± 0.24	37
DMPG	8.06 ± 0.08	3.66 ± 0.01	45	2.87 ± 0.04	36
DMPC + DMPG	79.24 ± 0.68	34.73 ± 0.26	44	28.85 ± 0.27	36
Cholesterol	57.08 ± 0.45	50.44 ± 2.01	88	5.45 ± 0.02	10
QS21	1.13 ± 0.01	1.08 ± 0.06	96	0.08 ± 0.02	7
MPLA	2.18 ± 0.05	2.04 ± 0.02	94	0.22 ± 0.01	10

* 5 mL of ALFQ suspension containing 22.9 mM total phospholipids. The amount of each lipid component was determined by UPLC–MS/MS, and % recoveries were calculated relative to the original uncentrifuged ALFQ. An unpaired *t*-test with Welch’s correction was employed to compare the lipid components (by weight) of GUVs (pellet) with SUVs (supernatant). The individual lipid components of GUVs were all statistically different (*p* < 0.005) compared with those of SUVs.

**Table 3 pharmaceutics-15-02212-t003:** Zeta potentials of liposomes in SPBS (pH = 6.1).

Liposomes	Zeta Potential (mV)
ALF55	−10.50 ± 0.52
ALFQ	−10.60 ± 0.62
Pellet (GUV)	−11.43 ± 1.15
Supernatant (SUV)	−8.38 ± 0.97

## Data Availability

All data are available from the authors.

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
