# Peer review of "QS21-Initiated Fusion of Liposomal Small Unilamellar Vesicles to Form ALFQ Results in Concentration of Most of the Monophosphoryl Lipid A, QS21, and Cholesterol in Giant Unilamellar Vesicles"

_pharmaceutics, 2023, doi:10.3390/pharmaceutics15092212_

Round 1
Reviewer 1 Report
Regarding the manuscript (pharmaceutics-2529554) entitled:
“QS21-Initiated Fusion of Liposomal Small Unilamellar Vesicles to Form ALFQ Results in Nearly Exclusive Concentration of Monophosphoryl Lipid A and QS21, and Most of the Cholesterol, in Giant Unilamellar Vesicles”
Comments to the Author
General comment
The manuscript describes the preparation of Army Liposome Formulation with QS21 (ALFQ). I have some few comments to be considered before publication:
1. Abstract: study objectives are missing
2. Introduction: more details about GUV, ALFQ with examples and preclinical and clinical studies should be discussed.
3. 2.2 ALFQ: Complete details about preparation method and optimization should be mentioned.
4. 2.3 AS01-like liposomes: Complete details about preparation method and optimization should be mentioned.
5. TEM, EE% should be conducted.
6. Stability study should be conducted in different conditions.
7. No preclinical application or clinical application were studied to show the efficiency of the developed system.
Reviewer 2 Report
This manuscript describes the characterisation of unilamellar vesicles within a lipid formulation that is known to have contained nanoscale vesicles, but further formulation characterisation studies have now revealed the existence of giant unilamellar vesicles within the mixture. These GUVs have been isolated, various experiments have been undertaken to investigate the reason they are present in the formulation, and potential explanations for their formation have been given.
The investigation into the formation of the GUVs is thorough and is described fully within the manuscript. However, the significance of this discovery is hidden somewhat, and the authors seem to have assumed the reader will know why GUVs are of interest, for example in targeted vaccine delivery. There is mention of their interest in the introductory paragraph and also towards the end of the paper, but I expected the authors to be more explicit regarding the use of the GUVs in the formulation of new modalities.
Overall, I think this is a very useful piece of research and provides an insight into the formation of GUVs, and the study is well presented, and I am happy to recommend publication after the authors have given a clear indication of the significance of this work and relevance to new modalities and formulation of APIs.
Reviewer 3 Report
This work is particularly interesting, since it links the biophysical profile of liposomal adjuvants with particle size, structure and possibly morphology. This however is not clearly emphasized in the conclusion part and it probably should. Additionally:
-Title is a bit complex.
-It would be preferable to avoid abbreviations in the title.
-In addition, all abbreviations should be explained in the manuscript.
-Why did the authors not measure the zeta potential by electrophoretic light scattering? This is an indicator of nanoparticle surface properties, system stability and biological behavior. If z-pot is already known, at least it can be mentioned and introduced into the concept of interactions between the components.
-It is this reviewer's belief that the authors should point out the ratio between QS21 and the rest of the membrane components (the most crucial for binding, ie cholesterol?) in the isolated GUVs, in order to later comment on the possible distribution (and conformation?) of the saponin on the particles.
-Since the weight values in Table 1 are the sum of individual components from Table 2, this should be clarified.
-Are all the self-citations necessary?
-Statistical analysis can confirm the observed differences.
Language is generally fine. Minor details such as Line 224 "1um" can be discovered after one final thorough reading.
Round 2
Reviewer 1 Report
NO COMMENTS